# Protocol for the feasibility and acceptability of a brief routine weight management intervention for postnatal women embedded within the national child immunisation programme: randomised controlled cluster feasibility trial with nested qualitative study (PIMMS-WL)

Helen M Parretti,[1] Natalie J Ives,[2] Sarah Tearne,[2] Alexandra Vince,[2] Sheila M Greenfield,[1] Kate Jolly,[1] Susan A Jebb,[3] Emma Frew,[1] Lucy Yardley,[4,5] Paul Little,[6] Ruth V Pritchett,[1] Amanda Daley  [7]

For numbered affiliations see end of article.

**Correspondence to**
Professor Amanda Daley;
a.daley@lboro.ac.uk

## ABSTRACT

**Introduction** On average women retain 5 to 9 kg 1 year after giving birth which can increase the risk of later obesity and chronic diseases. Some previous trials in this population have been effective in reducing weight, but are too intensive and costly to deliver at scale. There is a need for low-cost interventions to facilitate weight loss in this population.

**Methods and analysis** The primary aim is to assess the feasibility of delivering a weight management intervention for overweight/obese postnatal women within child immunisation appointments. We will conduct a randomised controlled cluster feasibility trial with a nested qualitative study to assess study recruitment and acceptability of the intervention. General practitioner practice (cluster) will be the unit of randomisation, with practices randomised to offer usual care plus the intervention or usual care only. Eighty women will be recruited. The intervention group will be offered brief support that encourages self-management of weight when attending child immunisation appointments. Practice nurses will encourage women to weigh themselves weekly and record this, and to make healthy lifestyle choices through using an online weight management programme. Women will be advised to aim for 0.5 to 1 kg/week weight loss. At each child immunisation the nurse will assess progress by weighing women. The comparator group will receive a healthy lifestyle leaflet. Data on weight, body fat, depression, anxiety, body image, eating behaviours and physical activity will be collected at baseline and follow-up. Women and nurses will be interviewed to ascertain their views about the intervention. The decision to proceed to the phase III trial will be based on prespecified stop-go criteria.

**Ethics and dissemination** Data will be stored securely at the University of Birmingham. Results will be disseminated

### Strengths and limitations of this study

► This is the first study to assess the feasibility of embedding a weight management intervention for postnatal women within usual care contacts in primary care.
► This study uses mixed methods and will draw on both quantitative and qualitative data to assess the feasibility and acceptability of the intervention.
► The intervention is based on self-regulation theory with external accountability.
► Rather than relying on self-reports, the study uses objective recordings from real time weighing scales to assess the frequency of self-weighing.

through academic publications and presentations and will inform a possible phase III trial. The National Research Ethics Committee approved the study protocol.

**Trial registration number** ISRCTN12209332

## INTRODUCTION

Pregnancy is associated with significant weight gain and many women report pregnancy as the critical point for the onset of excess weight.[1 2] The postnatal period is characterised not only by weight retention, but also susceptibility to further weight gain, which can significantly increase the risk of later obesity and serious chronic diseases.[3–5] There is also an association between postnatal weight retention and poor mental health.[6–8] It is estimated that on average women gain

about 14 to 15 kg during pregnancy and at 1 year after birth 5 to 9 kg is retained.[2 3 9 10] Of women who are overweight prior to conception, 44% are living with obesity by 1 year after giving birth, while 97% of women who are living with obesity prior to pregnancy remain so at 1 year postnatally.[5] Therefore, there is a need to intervene routinely in the postnatal period, to help women manage their weight after having a baby and minimise long-term health risks.

Several systematic reviews have identified the need for high quality trials on how to best help women to lose weight after having a baby.[11–16] Within these reviews, the vast majority of studies have recruited small sample sizes and evaluated intensive physical activity and/or dietary interventions delivered by specialists; while these types of interventions can be effective, they cannot be delivered to all the women who give birth each year in the UK, of whom nearly two-thirds will be living with overweight.[17] A recent systematic review of reviews of lifestyle interventions for postnatal weight management highlighted the lack of studies that have tested an intervention embedded within routine healthcare.[16]

### Self-management strategies and weight loss

A systematic review of randomised controlled trials (RCTs) to determine the effectiveness of self-help interventions for weight loss,[18] has shown that self-help interventions can lead to small, but significantly greater weight loss than unsupported attempts to lose weight at 6 months compared with minimal interventions. In the small group of studies providing data at 12 months, weight loss was no longer significant, suggesting that self-help interventions on their own may not be useful for sustaining long-term weight loss, and that additional components, within weight loss interventions are required. Nonetheless, given the potential scalability and relatively low cost of this type of intervention, self-help programmes may be a useful component within a broader intervention to treat those who are overweight.

Self-monitoring of weight by regular weighing is common to many self-management interventions. The potential efficacy of regular weighing is based on the principles of self-regulation theory.[19 20] Self-regulation theory describes three distinct stages; self-monitoring, self-evaluation and self-reinforcement. Self-monitoring is a method of systematic self-observation, periodical measurement and recording of target behaviours with the goal of increasing self-awareness. The awareness fostered during self-monitoring is an essential initial step in promoting and sustaining behaviour change. Strong evidence supports the role of self-monitoring as an effective strategy in the health behaviour change process. Reviews by Michie *et al* of effective behavioural techniques for healthy eating, physical activity and reduction of alcohol consumption concluded that self-monitoring was effective alone, but when combined with other techniques the effect size nearly doubled.[21 22] In a systematic review of RCTs to examine the effectiveness of self-weighing

as a strategy for weight loss, multi-component interventions including self-weighing compared with no/minimal control also resulted in mean differences of −3.7 kg (95% CI −4.6 to −2.9).[23]

### Accountability/audit and feedback

There has been some evidence to suggest that adding accountability to a self-weighing programme increases weight loss.[23] In a review by Madigan *et al*, in 13 of the included trials, the intervention group asked to weigh themselves knew that they were accountable to a therapist/researcher, while this was not the case in two trials. There was a significant difference in mean weight loss between intervention and control for those trials with accountability compared with those without (−4.0 kg, 95% CI −5.0 to −3.0 kg vs −2.3 kg, 95% CI −3.1 to −1.5 kg; p=0.007).[23] Participants in group weight loss programmes often report that it is the weekly weigh-in that is the most salient component of the programme, providing external accountability and keeping them committed to their diet and physical activity plan.[24] Based on this evidence, it can be hypothesised that including accountability/audit to self-help and self-monitoring interventions could further facilitate weight loss.

### Internet-based weight management interventions

The widespread use of technology has led to an interest in digital health interventions, including for weight management. Systematic reviews of eHealth or web-based interventions for weight loss and/or maintenance in adults have found that these interventions can result in modest weight loss in individuals with overweight or obesity.[25 26] One such digital interventions is the POWeR (Positive Online Weight Reduction) online weight management programme.[27] The POWeR programme was designed for use with minimal additional support and was developed with extensive user feedback; the development process and content of the POWeR intervention is described elsewhere.[28 29] The POWeR programme has been shown to result in significant weight loss at 12 months in primary care patients with overweight when combined with brief nurse support.[27] It focusses principally on fostering users' self-regulation skills for autonomously self-managing their weight, rather than providing detailed dietetic advice. A recent systematic review concluded that internet-based weight management interventions for postnatal mothers appear to be beneficial, and survey study results have reported that irrespective of age or socioeconomic status, 90.5% of new mothers use the internet as a source of information with weight loss as the fourth most searched topic.[20–31] Therefore, based on this evidence it may be hypothesised that including an online intervention may be beneficial for postnatal weight loss.

### Study rationale

There is a need to support women to lose excess weight in the postnatal period. Given the large numbers of women who give birth each year either with pre-existing

overweight or obesity, or who gain excess weight during this period, an intervention is needed that can be delivered at scale without significant implications for a healthcare workforce. In addition, the National Institute for Health and Care Excellence (NICE) has highlighted the low quality of previous research as a limitation to developing clinical guidance in this area.[32] Here we investigate the feasibility of a postnatal weight management intervention embedded within pre-existing healthcare contacts. This pragmatic intervention is based on existing evidence for the effectiveness of both self-monitoring and an online programme for weight management (POWeR).[19 27]

## Aims and objectives

The primary objective is to produce evidence that a large-scale phase III cluster RCT of a weight management intervention embedded in the national child immunisation programme[33] would be acceptable to women and feasible to conduct. Detailed objectives are listed in online supplementary table 1. This protocol is reported according to the Standard Protocol Items Recommendations for Interventional Trials guidelines.[34]

## METHODS AND ANALYSIS
### Trial design

PIMMS-WL is a feasibility cluster RCT to assess the feasibility and acceptability of a weight management intervention, based on self-regulation theory, in women who have recently given birth.[20] This will be assessed through a composite of quantitative and qualitative data. Key stages of the study are shown in figure 1.

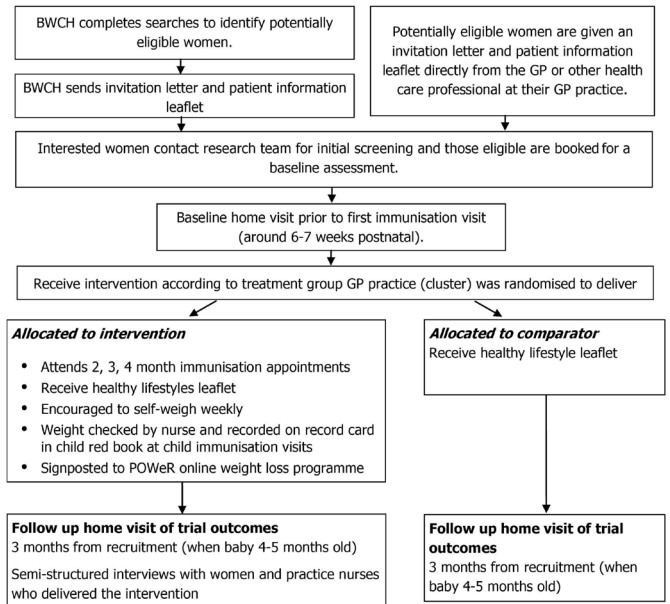

**Figure 1** Study flow diagram. BWCH, Birmingham Women's and Children's NHS Foundation Trust; GP, general practitioner; NHS, National Health Service.

## Trial setting

This study will take place in Birmingham, UK, between 01 September 2017 and 31 October 2019. The initial identification of women will take place within Birmingham Women and Children's NHS Foundation Trust and within participating generalpractitioner (GP) practices, and the trial will take place within GP practices around the West Midlands. The baseline and follow-up assessments will be completed in participants' homes.

## Study participants

Women will be eligible if they are aged ≥18 years, are at least 4 weeks postnatal, have a body mass index (BMI) ≥25 kg/m² at the time of recruitment, are able to provide written informed consent and do not meet any of the exclusion criteria (online supplementary table 2). Given that the intervention is embedded in immunisation appointments to be eligible, women need to have not yet attended their child's first immunisation appointment and to be planning to have their child immunised.

## Consent

Written informed consent into the PIMMS-WL trial will be a two-stage process conducted by a researcher at the baseline home visit (see online supplementary appendix 1 for Informed Consent Forms). First, written informed consent will be sought for screening to confirm all eligibility criteria are met. Once all eligibility criteria have been confirmed, participants will be invited to give their written informed consent to be enrolled into the RCT.

## Sample size and recruitment

This is a feasibility trial therefore a formal sample size calculation has not been conducted. The trial is not designed or powered to detect a statistically significant difference in effectiveness between the two treatment groups.

We will recruit 10 to 12 GP practices (clusters) across Birmingham, UK, and the list of participating practices will be held by Birmingham Clinical Trials Unit (BCTU). We will then aim to recruit 80 participants from these GP practices over 8 months. Potentially eligible participants will be identified initially from medical records at Birmingham Women's and Children's NHS Foundation Trust (BWCH). Initial screening of medical records by BWCH will confirm the women is aged ≥18 years, has given birth at least 4 weeks previously and is registered at a participating GP practice. We will exclude mothers whose babies have died or have been removed from their care.

A trial invitation letter and participant information sheet (PIS) will be mailed to potentially eligible women by BWCH, which includes a telephone number to call or a reply slip to return if they are interested in participating. Interested women will undergo telephone screening for eligibility. For women who fulfil the initial screening criteria and are still interested in taking part in the trial, a baseline home visit with a researcher will be organised.

Some additional recruitment strategies may also be used if required:

▶ Posters and flyers advertising the study will be displayed in waiting rooms at participating GP practices.
▶ Participants may be informed about the trial directly at appointments with the GP or other healthcare professional post-delivery and prior to the 2-month immunisation appointment in participating GP practices.
▶ A researcher may be available at the GP practice on specific clinic dates. The PIMMS-WL researcher will provide potential participants with the letter of invitation and PIS directly either to read at the GP practice or for the participant to take home.

If, after having read the letter of invitation and PIS, a woman is interested in participating, they will contact the PIMMS-WL research team for telephone screening (or be screened by the PIMMS-WL researcher at the GP practice) and a baseline home visit will be arranged as described above.

### Randomisation

The unit of randomisation is the GP practice (cluster). GP practices will be randomised in a 1:1 ratio to provide the usual care plus a weight management intervention or usual care only (comparator group) using minimisation for GP practice list size (large: 6000 or more; small: under 6000 patients) and Index of Multiple Deprivation (IMD) rank score.[35] The IMD will be based on the postcode of the GP practice; the IMD rank score ranges from 1 to 32 844 and will be divided into tertiles of high, medium and low levels of deprivation. BCTU will create the randomisation list to allocate GP practices to the two trial groups.

### Trial intervention (usual care plus weight management group)

The intervention will be delivered within the national child immunisation programme to minimise costs and additional burden on the National Health Service (NHS) workforce and avoid the need for additional contacts with health professionals at this busy time in women's lives. In the first year, babies are routinely immunised at 2, 3, 4 and 12 months of age; we plan to embed the intervention within these pre-existing immunisation contacts. The intervention group will be offered brief support that encourages active self-management of weight in the postnatal period when they attend their practice to have their child immunised during the first year of life. The intervention involves motivation and support by nurses to encourage women in the postnatal period to make healthier lifestyle choices through self-monitoring of weight and signposting to an online weight management programme (POWeR).[27]

Women will be provided with weighing scales and asked to weigh themselves weekly. The intervention will be delivered until the third immunisation appointment. Women will be advised and encouraged to record their weight on a weight record card that will be attached to the child health record (the 'red book') and also within the POWeR online programme.[27] The 'red book' includes a record of infant examinations and investigations, growth and immunisation history.

### Weight loss goals

No clinical guidelines exist for rates of healthy weight loss for postnatal women, but for the general population NICE recommend 0.5 to 1 kg per week.[36] Women will be advised to aim for 0.5 to 1 kg/week weight loss until they have achieved a BMI less than 25 kg/m$^2$ and are no heavier than their self-reported pre-pregnant weight.

### Accountability

Practice nurses will weigh the women at each child immunisation visit and record this weight as a source of regular external accountability. They will not provide any weight management advice.

### Online weight loss programme (POWeR)

Nurses will signpost women to the POWeR online weight loss programme for support and assistance with goal setting, action planning and implementation of changes to their lifestyle. Information about breast feeding and weight loss (such as diet, physical activity and rate of recommended weight loss) will be added to the programme for the purpose of this trial. Women will be encouraged to use the website weekly to track their weight, set and review eating and physical activity goals and receive personalised advice.

### Training of practice nurses

All nurses who deliver child immunisations at intervention practices will be trained to deliver the intervention following a standard protocol. A training manual will be provided, with instructions for the tasks to be undertaken by nurses during immunisation appointments.

### Intervention fidelity

Two immunisation/intervention consultations per nurse will be audio recorded so that intervention fidelity can be assessed against an intervention checklist criteria, and the length of time taken to deliver the intervention monitored.

### Usual care group

Women enrolled into the trial and registered at GP practices in the usual care comparator group will receive brief written information about following a healthy lifestyle using the NHS Eatwell guide leaflet.[37]

### Masking

It is not possible to mask participants or those providing the intervention to group allocation. In this feasibility study, it is also not possible to mask the outcome assessor, as the same member of the PIMMS-WL research team will need to undertake both the baseline and follow-up home visit to collect data. We do not believe this will introduce bias, as the aim of this study is to assess the feasibility of undertaking a large phase III cluster RCT, and these outcomes are not affected by knowledge of group

allocation and the data relating to the feasibility outcomes are not collected during the home visits. The proposed primary outcome for the phase III trial is weight which is objectively measured using scales.

## Baseline assessment and follow-up
### Baseline visit
The baseline visit will occur between 6 to 7 weeks postnatally and before the first immunisation visit.

### Follow-up visit
A follow-up visit will take place 3 months post baseline. Women in the intervention group will be invited to participate in the qualitative study interview (see below). A £20 gift voucher will be given to participants as reimbursement for any inconvenience and expenses incurred.

## Patient and public involvement
Members of the CLAHRC-WM (Collaboration for Leadership in Applied Health Research and Care West Midlands) maternity patient and public involvement (PPI) group (PRIME) provided input into the development of this study. They gave advice on the relevance of the study question based on their experiences and their assessment of the burden of the intervention on potential participants. Two members of the public will be invited to join the Trial Steering Committee (TSC). The PPI group will meet three times a year and will provide input into the topic guide for the qualitative interviews and have input into and review all patient facing documents. They will review all dissemination materials and be involved in the dissemination of the trial results.

## Primary outcome
The primary outcome is the feasibility of undertaking a full-scale phase III cluster RCT, and will include assessment of the following:
► Whether the trial is appealing to women (via assessment of the recruitment rate to ensure a full phase III trial is feasible);
► Whether the intervention is acceptable;
► Whether the intervention has any adverse impact on infant immunisation rates;
► The number of women who complete the trial and complete the trial questionnaires.

The decision to proceed will be based on prespecified stop-go criteria (see online supplementary table 3).

## Outcome data
Outcome data will be collected at baseline and 3 months follow-up (except height (baseline only)). While this feasibility trial will not be powered to detect meaningful differences in outcome measures, it will give us the opportunity to ensure that there are no issues with the completion of these measures in a main trial. Height and weight will be measured at baseline to calculate BMI. Weight and percentage body fat will be assessed using a Tanita SC-240MA analyser using a standardised procedure (eg, light clothing). Demographic (eg, age, ethnicity,

socioeconomic status), lifestyle and pregnancy related information (eg, history of pregnancy related problems, mode of delivery), infant feeding practices and maternal sleeping patterns will be collected. Anxiety and depression will be assessed using the Hospital Anxiety and Depression Scale, body image using the Body Image State Scale, eating habits using the three-factor eating questionnaire and self-reported physical activity will be measured using the Pregnancy Physical Activity Questionnaire.[38–41]

At follow-up, self-weighing will be assessed using the Daily Self-Weighing Perceptions Questionnaire and weight control strategies using the Weight Control Strategies Scale.[42] We will also record whether the usual care group have accessed the POWeR website to assess any intervention contamination. To inform the design of the economic evaluation in the phase III trial, we will explore the acceptability of the ICECAP instrument,[43] a broader measure of health than the EQ-5D[44] that focusses on general well-being. This will be done by assessing rates of completion of the ICECAP, and incorporating questions about the acceptability of the ICECAP into the qualitative interviews.

### Eating behaviours (both groups)
Our hypothesis is that regular self-weighing and monitoring of weight leads to the development of self-regulation and conscious cognitive energy restraint.[20 45] Using the revised three-factor eating questionnaire,[41] we will examine if feedback from self-weighing has led to development of conscious cognitive energy restraint of eating. We will also measure uncontrolled eating and emotional eating.

### Weight control strategies (both groups)
Weight control strategies will be assessed at follow-up in both groups using the Weight Control Strategies Scale. As well as a total score, the domains of dietary choices, self-monitoring strategies, physical activity and psychological coping are measured.[42]

### Objective assessment of self-weighing/adherence (intervention group)
As an objective measure of compliance, the intervention group will be given weighing scales (BodyTrace, USA) that objectively record weight every time participants weigh themselves and then this information is remotely transmitted back to the research team at the University of Birmingham. These scales are only included as an objective process measure of adherence to self-weighing and we will not provide any feedback to participants, nor will we monitor changes in weight during the trial follow-up period.

### Weight record cards (intervention group)
The intervention group will complete weight record cards and we will collect these from participants as a measure of intervention implementation at the follow-up visit. The record cards will also tell us how much of the intervention was delivered by practice nurses per protocol.

### Perceptions of regular self-weighing (intervention group)

In the intervention group, we will use five items from Steinberg's Daily Self-Weighing Perceptions Questionnaire[46] to measure perceptions of regular monitoring of weight at follow-up.

### Use of the POWeR online programme (intervention group)

Using participants' unique usernames POWeR automatically records participants' usage of the website (eg, pages visited, when/for how long, goals and weights entered and dates).

### Attendance at immunisation appointments (both groups)

To ensure the intervention does not have an adverse impact on women attending the immunisation appointments all practices will be asked to provide data on all immunisation appointments attended by both groups. Any missed appointments will be investigated and a reason allocated by the GP surgery. We will also collect patient reported attendance at the immunisation appointments at the follow-up visit.

### Assessment of harms

We will not be collecting adverse events for this trial. Serious adverse events (SAEs) which are attributable to the trial intervention will be documented and reported from the date of consent into the PIMMS-WL study until 30 days after the 3-month follow-up visit. SAEs or deaths related to pre-existing medical conditions are not considered related to the trial intervention and are therefore excluded from notification as SAEs.

### Withdrawal

Participants will be aware that they can freely withdraw from the trial (or part of) at any time. Details of withdrawal will be clearly documented. Different types of withdrawal are defined as:

- ► The participant would like to withdraw from trial treatment but is willing to be followed up in accordance with the schedule of assessments.
- ► The participant would like to withdraw from trial treatment and is not willing to be followed up or for any further data to be collected.
- ► The participant wishes to withdraw completely and is not willing to have any of their data, including that already collected, to be used in any future trial analysis.

The details of withdrawal (date, reason and type of withdrawal) should be clearly documented in the source data.

If a recruited GP practice withdraws from participation in the trial prior to randomisation, they may be replaced by another practice. If a practice withdraws after randomisation, consideration will be given to replacing with another practice.

### Statistical analysis

A separate statistical analysis plan will provide a more comprehensive description of the planned analyses. A brief outline of the planned analyses in relation to the stop-go criteria is provided below. The data analysis for this feasibility trial will be descriptive and mainly focus on CI estimation, with no hypothesis testing performed.

The recruitment rate will be presented as a percentage based on the number of participants who took part in the trial divided by the target recruitment (n=80). BWCH will also provide data on the number of invitation letters sent, along with data on the age and ethnicity (in summary format) of the women who were sent an invitation letter.

The quantitative assessment of whether the intervention is acceptable will be based on the adherence of women to weekly self-weighing and use of the online POWeR programme. In the first instance, the objective recording of weight on the scales will be used to assess the frequency of self-weighing/adherence.

In addition, to check that the intervention has no adverse impact on infant immunisation rates, the proportion of babies who attended all three immunisation appointments will be reported for each practice (both groups) and the rate compared with the normal immunisation rate for the practice. Reasons for missed appointments will be summarised descriptively. Recruitment, adherence and immunisation rates will be summarised as proportions with 95% CIs.

### Qualitative study

We will use semi-structured interviews after trial follow-up to explore the views of women about the intervention and experiences of the practice nurses delivering the intervention.[47] Topic guides will be informed by existing literature.

We will purposively sample up to 15 women (eg, age, ethnicity, socioeconomic status and BMI category at enrolment) and 8 to 10 practice nurses and ask them about their experiences of the trial, to allow for theme saturation to be reached.[48 49] Interviews will be conducted using a topic guide which will explore the women's experiences of and attitudes to trial participation, their understanding and rating of the importance of postnatal weight management, strategies used for weight management and their use of the online weight programme. For practice nurses, we will explore their experiences of delivering the intervention, barriers and facilitators, the content and phrasing of discussions with women about weight and views about the approaches that work best and least well. The objectives of the qualitative study are summarised in online supplementary table 4.

It is anticipated that interviews will take place face-to-face, but it is possible for logistical reasons that some will need to take place by telephone. Interviews will be recorded and transcribed with the permission of participants and thematically analysed using the Framework Method.[50] Data management will be facilitated by the use of QSR NVivo 11.[51] A list of overall and individual themes for women and practice nurses will be compiled to allow for cross group/individual comparison. Data collection and analysis will be iterative. All digitally recorded data that is collected during interviews will be recorded on an encrypted audio recording device. The transcriptions will be anonymised.

## Decision to progress to the phase III trial

For the phase III trial to take place there needs to be evidence from this feasibility trial of meeting prespecified stop-go rules. The trial is too small to include meaningful and sensitive stop-go criteria regarding the impact of the intervention on immunisation rates. However, we will check that the intervention has not adversely affected usual immunisation rates at each GP practice (as described above). The criteria to proceed to the phase III trial will be based on three criteria: recruitment rate, adherence to weekly self-weighing and registration with the online weight loss programme (POWeR) using the traffic light system shown in online supplementary table 3.

## Data collection and management

All data will be entered onto the PIMMS-WL trial database. All trial paperwork will be stored in lockable filing cabinets in a secure, swipe access part of the University of Birmingham. Password protected electronic databases, on secure University of Birmingham servers for trial data will have access limited to BCTU members of staff working on the trial. Further trial information on data management procedures is available on request via BCTU including data collection forms.

## ETHICS AND DISSEMINATION

The study will be conducted according to the principles of the Declaration of Helsinki and in accordance with other relevant national guidelines, regulations and acts.

## Monitoring and oversight

The trial will form part of a portfolio of studies hosted and managed by BCTU. The University of Birmingham holds the relevant insurance for this study and is the nominated sponsor for this study.

As this is a feasibility trial, we do not propose to hold a Data Monitoring and Ethics Committee. A TSC will provide overall supervision and oversight of the trial and ensure it is conducted in accordance with the principles of good clinical practice and relevant regulations. The TSC agreed the trial protocol and will agree to any protocol amendments. Submission of amendments and communication to the relevant stakeholders will be coordinated by BCTU. The TSC will include members who are independent of the investigators, their employing organisations, funders and sponsors. The TSC will monitor trial progress and conduct and advise on scientific credibility. An explicit purpose of the TSC is to monitor the immunisation uptake rates at the GP practices.

## Dissemination

A lay summary of the study is available on the National Institute for Health Research website (https://bepartofresearch.nihr.ac.uk/trial-details/trial-detail?trialId=17314&location=&distance=). Final results of this feasibility study will be made publicly available through publication in peer-reviewed journals and presented at relevant conferences.

## DISCUSSION

The postnatal period is characterised not only by weight retention, but also by susceptibility to further weight gain. Most trials in this population include small sample sizes and/or evaluated intensive weight loss interventions. These types of interventions cannot be offered to all women who give birth in the UK every year. This study will generate evidence as to whether a full phase III trial to test the effectiveness of this novel postnatal weight management intervention embedded within existing primary care contacts is feasible. Currently there is no national guidance for health professionals regarding postnatal weight management and the full trial testing the effectiveness of this intervention would have the potential to influence policy and future national guidelines.

**Author affiliations**
[1]Institute of Applied Health Research, University of Birmingham, Birmingham, UK
[2]Birmingham Clinical Trials Unit, Public Health Building, University of Birmingham, Birmingham, UK
[3]Nuffield Department of Primary Care Health Sciences, University of Oxford, Oxford, UK
[4]Department of Psychology, University of Southampton, Southampton, UK
[5]Primary Care and Population Sciences, Aldermoor Health Centre, Aldermoor Close, Southampton, UK
[6]School of Sport, Exercise and Health Sciences, Loughborough University, Loughborough, UK
[7]School of Sport, Exercise and Health Sciences, Loughborough University, Loughborough, UK

**Contributors** AJD developed the original idea for the study along with HMP, KJ, SJ and RP. LY and PL contributed to the protocol in relation to digital technology. HMP, AJD, ST and NI drafted the paper with input from all authors. EF was responsible for writing the health economics aspects of the protocol. SG was involved in developing the qualitative component of the study protocol. NI was responsible for the statistical methodology within the protocol. ST and AV further contributed to the development of the protocol after funding was awarded. All authors read, commented and approved the final manuscript.

**Funding** This work is supported by the National Institute of Health Research (NIHR) Health Technology Assessment programme grant number 15/184/14. HMP was funded by a National Institute for Health Research Academic Clinical Lectureship.

**Competing interests** None declared.

**Patient consent for publication** Not required.

**Ethics approval** This manuscript is based on Protocol V9.0 dated 16 June 2019—submitted and approved by the West Midlands and Black Country NHS Research Ethics Committee (17/WM/0399, approval date: 17 June 2019).

**Provenance and peer review** Not commissioned; externally peer reviewed.

**Data availability statement** Data sharing not applicable as no data sets generated and/or analysed for this study. Not applicable as no data has been generated from this publication.

**ORCID iD**
Amanda Daley http://orcid.org/0000-0002-4866-8726

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
