## [Reviewer comments · BMJ Open]

ARTICLE DETAILS

TITLE (PROVISIONAL)	Protocol for the feasibility and acceptability of a brief routine weight management intervention for postnatal women embedded within the national child immunisation programme: randomised controlled cluster feasibility trial with nested qualitative study (PIMMS-WL)
AUTHORS	Parretti, Helen; Ives, Natalie J; Tearne, Sarah; Vince, Alexandra; Greenfield, Sheila; Jolly, Kate; Jebb, Susan A; Frew, Emma; Yardley, Lucy; Little, Paul; Pritchett, Ruth V; Daley, Amanda

VERSION 1 – REVIEW

REVIEWER	Maria D'Amico Queen Mary University of London, United Kingdom
REVIEW RETURNED	05-Nov-2019

GENERAL COMMENTS	This protocol is very well written, the study design and procedures outlined are sound. I recommend that the manuscript is accepted with minor revisions. Please see a few points that came up for me while reading this protocol for you and the team to consider and respond. Page 8 of 32, lines 8-9: Typo as sentence begins with a "G" Page 17 of 32, lines, 25-27: Insert approval date for ethics committee >Can you clarify why the feasibility study has a 3 month follow up period instead of 12 months? I understand that the outcome data is being collected for the routine immunisation appointments at 2, 3, 4, month but not for the routine appointment at 12 month, why is this? In the full scale trial will the primary outcome (effectiveness) be collected at 3 months and/or 12 months? > This study is being undertaken from Birmingham hospital and GP practices in the West midlands. This may limit generalisability due to variability in secondary and primary care pathways between regions. This is important to consider as the feasibility findings will inform the development of a full scale effectiveness trial across the UK. > As this is a pragmatic trial, embedding a brief intervention into a national immunisation programme, I wondered if there are any retention strategies used to reduce loss to follow-up? The protocol mentioned that attendance/adherence will be monitored and will inform the STOP-GO criteria, but I couldn't get a sense if there
--

	were any efforts or procedures involved in maximising retention. For example, reminders of the nurse contact by the researcher or rescheduling of nurse contact if a mother leaves after her child's immunisation. I hope my comments make sense, and apologies if I have misunderstood the study in anyway. Good luck with this publication, it was very interesting and I am looking forward to the study results in the future!
--	--

REVIEWER	Reeta Lamminpää University of Eastern Finland
REVIEW RETURNED	19-Nov-2019

GENERAL COMMENTS	Interesting paper and well justified. There is a need for this kind of low-cost interventions in this group of women.
---

VERSION 1 – AUTHOR RESPONSE

Reviewer: 1

Reviewer Name: Maria D'Amico

Page 8 of 32, lines 8-9: Typo as sentence begins with a "G"

Thank you, this has been corrected to "Given"

Page 17 of 32, lines, 25-27: Insert approval date for ethics committee

Thank you, the date for ethics approval has been inserted on page 16. We have also corrected the date of protocol V9.0, apologies for the error (pages 14 and 16).

Can you clarify why the feasibility study has a 3 month follow up period instead of 12 months? I understand that the outcome data is being collected for the routine immunisation appointments at 2, 3, 4, month but not for the routine appointment at 12 month, why is this? In the full scale trial will the primary outcome (effectiveness) be collected at 3 months and/or 12 months?

In the full trial the primary outcome would be effectiveness at 12 months. The aim of this study is to test the feasibility of the trial methods and the feasibility and acceptability of the intervention. It was considered that 3 months follow-up was sufficient time to test these fully. In addition, the costs of running the feasibility study to 12 months would have been prohibitive.

This study is being undertaken from Birmingham hospital and GP practices in the West midlands. This may limit generalisability due to variability in secondary and primary care pathways between regions. This is important to consider as the feasibility findings will inform the development of a full scale effectiveness trial across the UK.

Thank you. We are only using secondary care to identify women who have recently given birth, for recruitment to the trial. No other secondary-primary care pathway is relevant to the trial. All hospitals keep a record of deliveries so this recruitment method should be applicable to all hospitals where women give birth. There is some variation within GP practices with regards to how postnatal checks and immunisation appointments are organised in the practice. However, we will capture a good representation of this variability within the practices we will recruit around the West Midlands. We have no reason to believe this would be different in other areas of the country.

As this is a pragmatic trial, embedding a brief intervention into a national immunisation programme, I wondered if there are any retention strategies used to reduce loss to follow-up? The protocol mentioned that attendance/adherence will be monitored and will inform the STOP-GO criteria, but I couldn't get a sense if there were any efforts or procedures involved in maximising retention. For

example, reminders of the nurse contact by the researcher or rescheduling of nurse contact if a mother leaves after her child's immunisation.

Thank you, we have included strategies to reduce loss to follow up, for example we will be conducting home visits for data collection at baseline and follow-up and also providing a £20 financial incentive at completion of the follow-up visits.

Reviewer: 2

Reviewer Name: Reeta Lamminpää

No revisions requested/comments